# Ovine and Caprine Strains of *Corynebacterium pseudotuberculosis* on Czech Farms—A Comparative Study

**DOI:** 10.3390/microorganisms12050875

**Published:** 2024-04-27

**Authors:** Jirina Markova, Denisa Langova, Vladimir Babak, Iveta Kostovova

**Affiliations:** Department of Microbiology and Antimicrobial Resistance, Veterinary Research Institute, 62100 Brno, Czech Republic; denisa.langova123@gmail.com (D.L.); vladimir.babak@vri.cz (V.B.); iveta.kostovova@gmail.com (I.K.)

**Keywords:** CLA, *Corynebacterium pseudotuberculosis*, small ruminants, antibiotic susceptibility, disinfectant agents, comparative genomics, virulence factors, CAZymes

## Abstract

Caseous lymphadenitis (CLA) is a worldwide disease of small ruminants caused by *Corynebacterium pseudotuberculosis*, a facultative intracellular pathogen that is able to survive and multiply in certain white blood cells of the host. In this study, 33 strains of *C. pseudotuberculosis* were isolated from sheep and goats suffering from CLA on nine farms in the Czech Republic. All these strains were tested for their antibiotic susceptibility, ability to form a biofilm and resistance to the effects of commonly used disinfectant agents. To better understand the virulence of *C. pseudotuberculosis*, the genomes of strains were sequenced and comparative genomic analysis was performed with another 123 genomes of the same species, including *ovis* and *equi* biovars, downloaded from the NCBI. The genetic determinants for the virulence factors responsible for adherence and virulence factors specialized for iron uptake and exotoxin phospholipase D were revealed in every analyzed genome. Carbohydrate-Active Enzymes were compared, revealing the presence of genetic determinants encoding exo-α-sialidase (GH33) and the CP40 protein in most of the analyzed genomes. Thirty-three Czech strains of *C. pseudotuberculosis* were identified as the biovar *ovis* on the basis of comparative genome analysis. All the compared genomes of the biovar *ovis* strains were highly similar regardless of their country of origin or host, reflecting their clonal behavior.

## 1. Introduction

The Gram-positive bacterium *Corynebacterium pseudotuberculosis* is one representative of the genus *Corynebacterium* belonging to the class *Actinobacteria* along with the genera *Mycobacterium*, *Nocardia* and *Rhodococcus*. This “CMNR group” includes bacterial species important for both veterinary and human medicine and is characterized by some common features such as high GC content and the organization of the cell wall [1,2]. *C. pseudotuberculosis* is divided into two biovars—biovar *ovis* and biovar *equi*—based on the differences in the results of the nitrate reduction test [3]. Although these biovars do not show strict host adaptation, the results of some studies and primarily sequencing data deposited at the National Center for Biotechnology Information (NCBI) show that the biovar *ovis* is mainly isolated from infections in small ruminants and the biovar *equi* from horses and large ruminants, e.g., dairy cattle [4,5].

In goats and sheep, infection with *C. pseudotuberculosis* biovar *ovis* causes caseous lymphadenitis (CLA) characterized by the formation of encapsulated inflammatory foci in the superficial lymph nodes or in the subcutaneous tissue. The internal form of CLA causes inflammatory changes in the organs and lymph nodes inside the animal’s body [6]. In the Czech Republic, precise data on the prevalence of CLA on individual sheep and goat farms are still lacking; however, the incidence of abscesses associated with the bacterium *C. pseudotuberculosis* is recorded [7,8]. The spread of CLA on Czech small ruminant farms represents an economic burden for breeders, related primarily to the culling of positive animals or the regular diagnostic tests needed to detect CLA-positive animals. In addition to eliminating CLA-positive animals, key biosecurity points need to be addressed, including disinfection, which is important to achieve a reduction in infection pressure. In order to survive in external environmental conditions, some bacteria have adapted and form a bacterial biofilm-forming population. Biofilms formed by *C. pseudotuberculosis* have been investigated in terms of morphology and biochemical composition and have already been subjected to some phenotypic and comparative proteomic analyses [9,10,11,12].

Modern methods of genome analysis and the availability of whole-genome sequences of *C. pseudotuberculosis* strains from different countries and hosts provide an increasingly high-quality basis for comparative genomic studies. Previously obtained data can be continuously expanded and refined to understand differences at the molecular level and to study evolutionary changes. Comparative genomic analysis in bacterial pathogens can help identify common or unique genes related to bacterial virulence, antimicrobial resistance and adaptation to environmental conditions, which can subsequently be targeted from a therapeutic and immunological perspective.

Our study combines genomic and phenotypic analyses of *C. pseudotuberculosis* strains obtained from CLA-positive sheep and goats in the Czech Republic to better understand the pathogenesis and survival of these bacterial strains derived from natural infections. This study also aims to compare the occurrence of virulence-related traits with another *C. pseudotuberculosis* biovar *ovis* and biovar *equi* genomes and to perform pan-genomic analysis to determine phylogenetic relationships.

## 2. Materials and Methods

### 2.1. Strain Isolation and Cultivation

Samples from purulent lesions were collected on sheep and goat farms in the Czech Republic with suspected CLA in 2019 and 2020. The contents of abscesses associated with superficial lymph nodes or localized in the subcutaneous tissue and internal organs (liver, lungs) were cultured on Columbia agar plates with 5% ram’s blood (LabMediaServis, Jaromer, Czech Republic) at 37 °C in a controlled CO_2_ atmosphere incubator (5% CO_2_) for 24–48 h. Suspected colonies were subcultured under the same conditions and identified by MALDI-TOF (Matrix-Assisted Laser Desorption/Ionization-Time-of-Flight) mass spectrometry (Bruker Daltonics GmbH, Bremen, Germany). A total of 33 strains identified as *C. pseudotuberculosis* were obtained from nine different farms in the Czech Republic. These were stored at −70 °C in nutrient broth No. 2 (Oxoid, Basingstoke, UK) supplemented with 15% glycerol. For further analysis, strains were subsequently revived on Columbia agar under the conditions described above, subcultured twice and re-verified by MALDI-TOF.

### 2.2. Biofilm Formation

All 33 strains of *C. pseudotuberculosis* were inoculated into 10 mL of Tryptic Soy Broth (TSB) liquid medium (Oxoid, Basingstoke, UK) supplemented with TWEEN 80 (1 g/L) and glucose (10 g/L) to support biofilm formation. Inoculated cultures were incubated at 37 °C in a controlled CO_2_ atmosphere incubator for 24 h. All samples were subsequently centrifuged (4816 g/5 min.) and washed with 5 mL of sterile saline solution (0.9% NaCl) to remove residual TWEEN 80 thoroughly. These samples were centrifuged, the cell pellet was re-suspended in 2 mL of TSB medium at a concentration of 1 × 10^8^ cells/mL and the suspension was pipetted into 96-well microplates at a volume of 200 µL per well. The microplates were incubated at 37 °C for 48 h in a controlled CO_2_ atmosphere incubator. Biofilm-forming *Staphylococcus epidermidis* CCM 7221 and biofilm-non-forming *S. epidermidis* ATCC 12228 were used as positive and negative control, respectively. After incubation, the biofilm was characterized by gentian violet staining [13] and the OD of each plate was measured at 570 nm using a Tecan microplate reader (Schoeller Instruments, Prague, Czech Republic). Each strain was tested in seven wells and the arithmetic mean of the OD values (ODI) was calculated. A classification based on the mean of the OD values for the negative control (ODC) was used for evaluation of biofilm formation (Appendix A Appendix A) [14,15].

### 2.3. Susceptibility of C. pseudotuberculosis Biofilm to Disinfection Solutions

Disinfectants belonging to the chemical group of quaternary ammonium compounds, oxidizing compounds, biguanides and alcohols were used to test the susceptibility of bacterial biofilm in microplates according to the European standard CSN EN 1656 [16]. These disinfectants were selected based on their use on the individual CLA-positive farms from which the *C. pseudotuberculosis* strains originated. *Staphylococcus aureus* ATCC 6538, *Streptococcus uberis* ATCC 19436, *Pseudomonas aeruginosa* ATCC 15442, *Escherichia coli* ATCC 10536, *Enterococcus hirae* ATCC 10541 and *Proteus vulgaris* CAPM 5087 were used as reference strains to verify disinfecting effects. A stock solution of each disinfectant was prepared depending on the type of disinfection agents and on the concentration range recommended by the manufacturer. Every disinfectant was diluted with either sterile distilled water or sterile tap water to imitate farm procedures. A prepared dilution series of disinfectants included benzalkonium chloride (≤0.0001–≥0.1%), sodium hypochlorite (≤0.001–≥1%), peracetic acid (≤0.00098–≥1%), chlorhexidine digluconate (≤0.00195–≥2%), polyvinylpyrrolidone (PVP) iodine (≤0.00977–≥10%) and ethanol (≤0.06836–≥70%). Each microplate contained one strain of *C. pseudotuberculosis*, the solution for control of the neutralizing effect, a positive control containing pure bacterial culture of the tested *C. pseudotuberculosis* strain without subsequent disinfection and TSB medium as a negative control. The recommended contact time of the disinfectants was 5 min for chlorhexidine digluconate and PVP iodine and 30 min for the other solutions. The minimum bactericidal concentration (MBC) was visually read as the lowest concentration of disinfectant that inhibited bacterial growth.

### 2.4. Strain Susceptibility to Antibiotics

Testing of the antibiotic susceptibility of *C. pseudotuberculosis* strains was performed according to the European Committee on Antimicrobial Susceptibility Testing (EUCAST) recommendations for the broth microdilution method using a medium for *Corynebacterium* spp. [17] and EUCAST reading guide [18]. The following antimicrobials were tested: benzylpenicillin (≤0.016–≥16 mg/L), vancomycin (≤0.032–≥32 mg/L), erythromycin (≤0.008–≥16 mg/L), clindamycin (≤0.016–≥16 mg/L), linezolid (≤0.032–≥32 mg/L), rifampicin (≤0.032–≥64 mg/L), trimethoprim/sulfamethoxazole (≤0.03/0.59–≥4/76 mg/L) and meropenem (≤0.016–≥16 mg/L). Antibiotics were selected based on EUCAST clinical breakpoints for *Corynebacterium* spp. and on the spectrum of antibiotic classes used on each sheep and goat farm. The microplates were incubated for 48 h at 36 °C in a controlled CO_2_ atmosphere and the minimum inhibitory concentration (MIC) as the lowest concentration of tested antibiotic that inhibited bacterial growth was visually read. Interpretation criteria were based on EUCAST Clinical Breakpoint Tables Version 12.0 [19] with microbiological cut-off values established for *Corynebacterium* spp., as well as on Clinical and Laboratory Standards Institute (CLSI) [20] categorization in the case of meropenem. Microplate quality control was performed using reference strain *Streptococcus pneumoniae* ATCC 49619.

### 2.5. Statistical Analysis

Statistical analyses were performed using the R-project software (version 4.1.2). *p*-values lower than 0.05 were considered statistically significant. In the case of MBC, log-transformed data were analyzed using PCA (principal component analysis; package vegan) and consequently using permutation analysis of variance (PERMANOVA; package vegan) at the level of all tested disinfectants. In addition, a PERMANOVA was performed for individual disinfectants. This method was also used to evaluate the MIC values at the level of all tested antibiotics and at the level of individual antibiotics (except for vancomycin and rifampicin due to zero variability). Fisher’s exact test (package stats) was performed for benzylpenicillin and meropenem to assess the significance of differences in the proportions of *C. pseudotuberculosis* strains isolated from sheep and goats with resistance or intermediate resistance to these antibiotics.

### 2.6. Whole-Genome Sequencing and De Novo Assembly

Genomic DNA was isolated using a Quick-DNA^TM^ Fecal/Soil Microbe Microprep Kit according to the manufacturer’s instructions (Zymo Research, Irvine, CA, USA). Extracted genomic DNA was subjected to whole-genome sequencing (WGS) using Illumina paired-end NovaSeq sequencing at Eurofins genomics (https://eurofinsgenomics.eu, accessed on 23 January 2023). The raw reads were trimmed and filtered from low-quality reads using Trim Galore v.0.6.6 (www.bioinformatics.babraham.ac.uk, accessed on 20 February 2023) powered by Cutadapt v.0.6.6. The quality of filtered and trimmed reads was evaluated by MultiQC v.1.9 [21] and de novo genome assembly was performed with Unicycler v0.4.9b [22].

### 2.7. Strain Identification Using Average Nucleotide Identity

The resulting genomic sequences of our isolates and genomic sequences of *C. pseudotuberculosis* biovar *ovis* and biovar *equi* downloaded from the NCBI (Appendix A) were compared by average nucleotide identity (ANI) calculations using FastANI v 1.32 [23]. The genome sequence of the type strain (*C. pseudotuberculosis* ATCC 19410) was used as a reference.

### 2.8. Genome Annotation and Comparative Genome Analysis

All genomes of *C. pseudotuberculosis*, including the genomes of our isolates as well as those downloaded from the NCBI, were subjected to genome prediction and annotation using Prokka v.1.14.6 [24]. All databases used by Prokka have been updated as of December 2020. Prokka-generated protein sequences were searched against the Carbohydrate-Active Enzymes (CAZy) database using the run_dbcan4 tool [25], with all three substrate prediction approaches used. Only coding sequences (CDS) annotated by at least two of the run_dbcan4 tools were considered CAZymes. Functional annotation based on precomputed orthology assignments was performed by the EggNOG-mapper tool e-mapper v.2.1.6.-25-g1502c0F [26], with protein sequences searched against the EggNOG database (EggNogDB version 5.0.2) by the DIAMOND v.2.0.11 protein aligner [27]. Protein sequences generated by Prokka were also used for pan-genome calculation by ROARY v.3.13.0 with the parameter of blastp identity 95% [28]. Core gene alignment was used for the construction of a phylogenetic tree based on the General Time-Reversible model with a model Γ of rate heterogeneity (GTR + G) using the tool RAxML-NG [29]. Additionally, for better resolution of all used *C. pseudotuberculosis* biovar *ovis* strains, including those from the NCBI, a phylogenetic tree was constructed based on SNPs generated by the Parsnp tool. All phylogenetic trees were decorated in iTOL [30] and Inkscape.

### 2.9. Genomic Detection of Antimicrobial Resistance and Virulence Factors

Antimicrobial resistance (AMR) and virulence-associated genes were detected using Abricate v.1.0.1 software (https://github.com/tseemann/abricate, accessed on 16 October 2023) with the use of the following databases: Comprehensive Antibiotic Resistance Database (CARD) [31], ResFinder [32], Argannot [33], Megares [34], NCBI AMRFinderPlus [35] and the virulence database VFDB. The parameters used for AMR detection were minimum DNA identity 60% and minimum sequence coverage 80%.

### 2.10. Accession Numbers of Sequenced Genomes

Scaffold sequences of *C. pseudotuberculosis* were deposited in the GenBank database under the corresponding NCBI biosample accession numbers listed in Appendix A.

## 3. Results

### 3.1. Biofilm Formation and Susceptibility to Disinfection Solutions

All 33 *C. pseudotuberculosis* strains were able to form a biofilm. The addition of 1% glucose to the culture medium resulted in the formation of strong or moderate biofilms, with 23 strains (69.7%) forming a strong biofilm and 10 strains (30.3%) forming a moderate biofilm. The concentrations of each disinfectant within their dilution series showed a sufficient bactericidal effect on all 33 strains of *C. pseudotuberculosis* and it was not necessary to use higher concentrations. The maximum values of MBC did not exceed the concentrations recommended by the disinfectant manufacturers and the MBCs were even lower for most strains (Table 1). MBC values of 0.5% for peracetic acid were measured for two strains forming a strong biofilm (CP-K12 and CP-K61) from two different farms. This MBC value lies in the middle of the range of effective concentrations (0.3–1%) established for peracetic acid by disinfectant manufacturers. The highest MBC value for PVP iodine was also measured for strain CP-K12 (MBC 10%), which is at the upper limit of the percentage concentration of PVP iodine contained in commercial products intended for topical application.

The output of the PCA for the MBC values, the scatter plot (Figure 1), shows a considerable dispersion of all values, as well as significant overlap in the confidence regions that correspond to the strains originating from sheep and goats. The PERMANOVA confirms what can be seen in Figure 1, i.e., there was no statistically significant difference between goat and sheep *C. pseudotuberculosis* strains at the level of all tested disinfectants. A partial statistically significant difference was demonstrated only for ethanol (*p* < 0.05; PERMANOVA), with statistically significantly higher MBC values for goat strains.

### 3.2. Corynebacterium pseudotuberculosis Susceptibility to Antibiotics

The MIC values of eight different antibiotics were obtained for 33 strains of *C. pseudotuberculosis* from sheep and goat farms in the Czech Republic (Table 2). According to EUCAST clinical breakpoint tables, benzylpenicillin resistance was detected in 24 of 33 (72.7%) strains originating from all nine tested farms with MICs ranging from 0.25 mg/L to 0.5 mg/L. The remaining nine strains had MIC values (0.125 mg/L) only one dilution above the established cut-off values. In the case of meropenem, ten strains (30.3%) of *C. pseudotuberculosis* from six farms were classified as having intermediate resistance with MIC values of 0.5 mg/L. Another twelve strains were only one dilution (MIC 0.25 mg/L) above the established cut-off values for the intermediate category. Resistance to other tested antibiotics was not observed.

The PERMANOVA for benzylpenicillin, erythromycin, clindamycin and meropenem showed no significant difference between the MIC values of goat and sheep strains (*p* > 0.05). A statistically significant difference was demonstrated in the case of linezolid (*p* < 0.05) and trimethoprim/sulfamethoxazole (*p* < 0.01), with the MIC values being statistically significantly higher in goat strains. Based on MIC values and interpretation criteria, there was no statistically significant difference between benzylpenicillin-resistant *C. pseudotuberculosis* strains from goats and those from sheep (*p* > 0.05; Fisher’s exact test). The same result was also obtained for strains categorized as intermediate to meropenem.

### 3.3. Genome Sequencing, Annotation and Strain Identification Using ANI Values

The genomes of all isolates were sequenced and assembled. The number of contigs representing the whole genome varied from five to nine, N50 values were in a range of 462,239–721,958 bp and L50 was represented by two contigs for every genome sequenced in this study. Also, regardless of biovar, the GC content within the genome corresponded to 52%.

Genome annotation performed for each genome, including those from the NCBI using Prokka, generated 2099–2387 CDS, 48–52 tRNAs and 6–7 rRNAs per genome.

Every strain sequenced in this study, as well as *C. pseudotuberculosis* genomes downloaded from the NCBI, was compared using average nucleotide identity values (ANI) using the FastANI method. In general, genomes that share ANI values greater than 95% belong to the same species [23]. The ANI values for each used genome in comparison with the genome of the type strain (*C. pseudotuberculosis* ATCC 19410) were higher than 98%. Therefore, all used strains belonged to the species *C. pseudotuberculosis*. Since the analyzed set of genomes contained genomes of the two known biovars of *C. pseudotuberculosis*, the difference between them was also reflected in slightly different ANI values. The ANI values in genomes belonging to the biovar *equi* were in the range 98.59–98.88%, whereas the ANI values in genomes belonging to the biovar *ovis* were slightly higher (99.84–100%) (Appendix A Appendix A). All *C. pseudotuberculosis* strains sequenced in this study had ANI values of 99.98% and were more similar to the ANI values of genomes belonging to the biovar *ovis* (Appendix A).

### 3.4. Pan-Genomic Analysis

The dataset for pan-genomic analysis consisted of 156 genomes of *C. pseudotuberculosis* biovar *ovis* and *equi*, including 33 genomes of *C. pseudotuberculosis* strains isolated in this study and 123 genomes downloaded from the NCBI. The used genomes of the *C. pseudotuberculosis* biovars *ovis* (n = 81) and *equi* (n = 42) originated from the following countries: unspecified country in Africa (n = 2, more detailed geographical information not specified), Argentina (ARG, n = 3), Australia (AUS, n = 2), Belgium (BEL, n = 2), Brazil (BRA, n = 49), Egypt (EGY, n = 14), France (FRA, n = 1), the United Kingdom of Great Britain and Northern Ireland (GBR, n = 2), Switzerland (CHE, n = 6), China (CHN, n = 2), Israel (ISR, n = 3), Kenya (KEN, n = 1), Mexico (MEX, n = 5), Norway (NOR, n = 2), Portugal (PRT, n = 2) and the United States of America (USA, n = 17), in addition to ten genomes of *C. pseudotuberculosis* of unknown geographical location and, finally, isolates from the Czech Republic sequenced in this study (CZE, n = 33).

Similarly to those in our study, genomes downloaded from the NCBI belonged in most cases to bacterial strains isolated from animal sources. The most frequent isolation sources were sheep (n = 59), goats (n = 47) and horses (n = 30). Other less frequent isolation sources were represented by cattle (n = 4), camels (n = 1), buffalo (n = 1), water buffalo (n = 10), wildebeest (n = 1), llama (n = 1) and also humans (n = 1). Based on isolation source data, it can be seen that the animals that can suffer from illness caused by both biovars include cattle and water buffalo. The only source of isolation of the *C. pseudotuberculosis* biovar *equi* was the horse, whereas the biovar *ovis* was isolated only from sheep and goats.

The pan-genome of the dataset was calculated by ROARY, which generated 3824 gene clusters. The pan-genome is represented by core genes (1552, 154 ≤ strains < 156), soft-core genes (124 genes in 148 ≤ strains < 154), shell genes (687 genes in 23 ≤ strains > 148) and cloud genes (1361 genes in < 23 strains). The number of new genes did not increase further with the addition of the genome into the pan-genome (Figure 2), indicating that the pan-genome defined in this study approached the general *C. pseudotuberculosis* pan-genome. To verify this finding, the data generated by ROARY were also processed by the R package Micropan, which calculated the α value of Heap’s law. The calculated α value (α = 2.0) was higher than one and the *C. pseudotuberculosis* pan-genome including both biovars, *equi* and *ovis*, can therefore be considered essentially closed.

The differences captured in the pan-genome matrix (Figure 3) belonged mainly to all genomes of biovar *equi* because, unlike the biovar *ovis*, the biovar *equi* possesses a nitrate locus that contains the narKGHJI operon which is composed of the molybdopterins *moeB*, *moaE*, *molB*, *molA*, *moeY*, *moaC*, *moeA* and *moa* [36]. Also, all genomes of the biovar *equi* contained the CrisprCas system (Class1, Subtype-I-E), which was absent in the genomes of the biovar *ovis*. Comparison of all *C. pseudotuberculosis* biovar *ovis* genomes did not show CDS encoding for any different biochemical pathways unique to only a group of some strains. The majority of differences between the genomes of this biovar were represented by hypothetical proteins with a random location in the genome and with an unknown function.

### 3.5. Phylogenetic Analysis Based on Core Genome Single-Nucleotide Polymorphism in Biovar Ovis Strains

The genome sequences of 81 *C. pseudotuberculosis* biovar *ovis* strains obtained from the NCBI and 33 strains of the same species and biovar sequenced in this study were processed using the tool Parsnp (Figure 4). According to Parsnp analysis, the core genome comprised 90.4% of the reference genome (strain ATCC 19410) with a total number of 7904 SNPs in the core genome. The phylogenetic tree can be divided into the main clades A and B and subclades B1 and B2 based on the number of SNPs within the core genome of the above-mentioned strains. The strains most closely related to the reference strain are within subclade B, which can be further divided into two other subclades, B1 and B2. The number of SNPs within the subclade B2 is 803–909, whereas the number of SNPs in B1 decreases to just 23–257 SNPs. The B1 subclade contains, besides other strains, 31 strains of the *C. pseudotuberculosis* biovar *ovis* sequenced in this study. All of these strains are very closely related to the reference strain ATCC 19410 and their numbers of SNPs in the core genome range from 122 to 230. Strains isolated in this study are extremely similar to strains isolated from sheep, goats, cattle and wildebeest in Argentina, Brazil, Australia, Israel, Mexico, Switzerland and the USA. Furthermore, strains isolated on the same farm clustered together in almost every case. The only exceptions were the strains CP_K12 and CP_K58, which differed from the rest of the Czech strains, showing closer relations to the strains within subclade B2. Additionally, the strains CP_K12 and CP_K58 displayed close relations with the strain PO222 4-1 isolated from a goat in Portugal and with the strain FRC41 from France which was isolated from a 12-year-old girl with necrotizing lymphadenitis [37].

### 3.6. Genomic Detection of Virulence Factors and Antimicrobial Resistance Determinants

All analyzed genomes of *C. pseudotuberculosis* bore CDS coding for virulence genes belonging to the following VF categories: adherence, nutritional/metabolic factors, regulation and toxin class. The detected virulence factors responsible for adherence belonged to the subcategories SpaA-type pili (*spaC*, *strA*), SpaD-type pili (*spaD*, *srtB* and *srtC*), SpaH-type pili (*spaI*), surface-anchored pilus protein (*sapA*) and CdiLAM (*aftB*, *emb* and *mptC*) (Appendix A). The VFs for nutritional/metabolic factors were represented by virulence factors specialized for iron uptake and belonging to the following VF subcategories: ABC transporters (*fagA*, *fagB*, *fagC* and *fagD*), ABC-type heme transporters (*hmuT*, *hmuU* and *hmuV*) and Ciu iron uptake and siderophore biosynthesis system (*ciuA*, *ciuB*, *ciuC*, *ciuD* and *ciuE*). In addition, CDS encoding for exotoxin phospholipase D and transcriptional regulator *dtxR* (diphtheria toxin repressor) were found in almost every genome in one copy (except for strain PA05). The only exception was the genome of PA05, which seems to have two copies of the CDS encoding for ABC transporters (*fagA*, *fagB*, *fagC* and *fagD*) and exotoxin phospholipase D. The diphtheria toxin precursor (*tox*) was also found, though only in seven biovar *equi* strain genomes isolated from buffalo and water buffalo in Egypt. The CDS encoding for the above-mentioned VF had the highest percentage identity with genes from *C. pseudotuberculosis*, except for the CDS encoding for the *dtxR* and *tox* genes which had high identity (coverage 100%, DNA identity 99–100% and 94%, respectively) with the same genes originating from *C. diphtheriae* NCTC 13129.

Regarding the detection of antimicrobial resistance according to analysis performed by the program Abricate (card and megares databases), CDS encoding for the rifampin-resistant beta subunit of RNA polymerase (*rpoB2*) and RNA-polymerase binding protein, which confers resistance to rifampin (*rbpA*), were detected. Although antimicrobial resistance determinants responsible for rifamycin resistance were found, the identity and coverage with reference sequences in the databases were relatively low. *rpoB2* showed only 98% coverage and 71% identity with rifamycin resistance gene *rpoB2* from *Nocardia* species (accession: AP006618.1:4835199-4838688) and *rbpA* showed only 82% coverage and 71% identity with the rifamycin resistance gene in both databases (accession: HQ203032, MEG_6047).

### 3.7. Detection of Carbohydrate Active Enzymes

The CAZyme families identified in both *C. pseudotuberculosis* biovars revealed the GH families involved in α-glucan metabolism, such as GH13, GH15 and GH77. The CAZymes responsible for α-glucan synthesis, which were all found in every genome of the analyzed dataset regardless of biovar, were represented by trehalose synthase (*treS*, GH13 subfamily 44), maltokinase (*pep2*, GT4), malosyltranferase (*glgE*, GH13 subfamily 3), 1,4-α-glucan branching enzyme (*glgB*, CBM48 + GH13 subfamily 9) [38,39] and GH77 corresponding to amylomaltase *malQ* with the ability to convert linear α-glucans into cyclic α-1,4-glucans [40].

The identified glycosyl transferases (GTs) were mainly connected to the synthesis of the corynebacteria-specific cell envelope consisting of peptidoglycan with attached linear galactan polymers and galactan polymers connected to branched arabinan polymers appended with mycolic acids [41]. The identified GTs indispensable for arabinogalactan synthesis were galactofuranosyltransferases *glfT1* and *glfT2* (GT2, EC 2.4.1.287, 2.4.1.288), arabinofuranosyltransferase *aftA* (GT85, EC 2.4.2.46) and arabinofuranosyltransferase aftB (GT89). The *aftB*, as well as mannosyltransferase *mptC* (GT87), which is involved in cell envelope synthesis, were also identified by Abricate as adherence virulence factors. The annotated carbohydrate esterases (CEs) code for mycolyltransferases and esterases, which are responsible for the incorporation of mycolic acids into the cell envelope [42].

Other identified CAZymes were not connected with glucan synthesis/hydrolysis or cell envelope and capsule formation, but with a possible impact on animal host infection, such as exo-α-sialidase (GH33) and GH18. In all analyzed *C. pseudotuberculosis* genomes, the annotated GH18 codes for CP40 (NCBI: KY041980.1) protein, which hydrolyses biantennary glycans on both human and ovine IgGs. Sequences in genomes of *C. pseudotuberculosis* biovar *ovis* identified as GH18 glycoside hydrolases exhibited 99–100% Blast sequence similarity to the CP40 CDS sequence. These enzymes were also found in genomes of biovar *equi* (except for strain 262), though their DNA Blast sequence similarity was lower than 90–92% compared to the biovar *ovis* [43].

The general CAZyme representation within all analyzed genomes of *C. pseudotuberculosis* revealed that the biovar *ovis* is extremely uniform with very few exceptions (Figure 5). Regarding our strains of the *C. pseudotuberculosis* biovar *ovis*, there were only three strains, all from farm C, that differed from the other biovar *ovis* strains. All isolates from farm C possessed two CDS encoding for CBM48 + GH13 subfamily 14 (CBM48 + GH13_14, pullulanase, EC 3.2.1.41, 3.2.1.68) instead of one. Comparison of the CAZyme profiles of the biovars *ovis* and *equi* showed that the majority of biovar *equi* strains (38 out of 42) encoded the second CDS encoding the enzyme glucan 1,4-α-glucosidase (EC 3.2.1.3, GH15). In general, the *C. pseudotuberculosis* biovar *equi* is more variable regarding CAZyme profiles compared to the biovar *ovis* strain.

## 4. Discussion

### 4.1. Pan-Genomic Analysis

The pan-genome analysis in our study investigates genomic variability at the species level to reveal variability among the *C. pseudotuberculosis* genomes of both biovars. Previously, a study by Soares et al. showed that the *C. pseudotuberculosis* biovar *ovis* exhibited a close relationship and clonal-like behavior [2], which is consistent with our results in which the genomes of the biovar *ovis* differentiated less than the genomes of the biovar *equi* strains. However, in the same study by Soares et al., pan-genomes were also calculated for each biovar separately (15 genomes of both biovars) and then extrapolated with resulting α values lower than one (Heap’s law α ≤ 1 = open pan-genome, Heap’s law α ≥ 1 = closed pan-genome), leading to open pan-genomes. In contrast, the larger dataset (156 genomes) including both biovars was used in our study for pan-genome calculation with the ROARY program, resulting in a closed pan-genome. The number of genomes may affect the openness/closeness of the pan-genome because the addition of a new strain genome may add new genes [2,44,45]. In general, a clonal character and a closed pan-genome is common to a pathogenic bacterial population already successful in colonizing animal tissues without any further need for adaptation to different environments [45].

The main differences among the analyzed genomes were recorded between the biovars *equi* and *ovis*. As in other studies, pan-genome analysis proved the absence of these CDS coding for genes connected to nitrate reduction in all genomes of the *C. pseudotuberculosis* biovar *ovis* strains as well as the absence of the CRISPR-Cas system [46,47]. Therefore, based on ANI values and the absence of CDS coding for genes connected to nitrate reduction, the strains sequenced in this study belong to the biovar *ovis*, which also corresponds to their isolation from sheep or goat lesions. The pan-genome analysis did not reveal any consistent gene difference between the genomes of goat and sheep isolates.

### 4.2. Phenotype Analyses

Phenotypic analyses of sensitivity in 33 *C. pseudotuberculosis* strains revealed susceptibility to selected disinfectants and most of the tested antibiotics. The ability to form a biofilm was also confirmed in all strains under laboratory conditions, which is consistent with the data reported in the literature [9,10,12]. As a result, the effect of disinfectants was therefore investigated when applied to biofilms of strong or moderate intensity having a greater potential for disinfectant resistance than the planktonic form of bacteria [9]. The selected disinfectants used on sheep and goat farms in the Czech Republic were effective in the recommended concentration ranges on the biofilms of all 33 tested strains. In a study conducted in Brazil with a consolidated biofilm of 398 *C. pseudotuberculosis* strains, the tested disinfectants did not achieve the same efficacy, although iodine and quaternary ammonium prevented biofilm formation in 33% and 28% of strains, respectively [10]. In our study, in two strains from two farms, MBC values for PVP iodine (strain CP-K12) and peracetic acid (strains CP-K12 and CP-K61) were at the upper limit or in the middle of the recommended concentration range, respectively. PVP iodine was the most commonly used disinfectant on the tested farms, applied topically to the surface wounds of sheep and goats or used for their rinsing with the possibility of dilution to lower concentrations (1% and 0.1%). The other groups of selected disinfectants (including peracetic acid) were mainly used on farms for disinfection in milking parlors and cheese factories, as most of the tested farms had herds on pasture all year round or did not use disinfection in stables.

According to the MIC values and EUCAST clinical breakpoints for *Corynebacterium* spp., antibiotic resistance of the *C. pseudotuberculosis* strains from our study was observed only in terms of benzylpenicillin (72.7%), and 30.3% of the strains were classified as intermediate to meropenem according to CLSI. Strains of *C. pseudotuberculosis* that are less sensitive or resistant to some β-lactam antibiotics have also been reported in samples originating from small ruminants in Spain [48], Kosovo [49] and Egypt [50]. Antibiotics for the treatment of CLA are used in the Czech Republic principally on small or hobby sheep and goat farms or in the case of valuable breeding animals. Therefore, the selected panel of antibiotics was designed to take into account the antibiotic groups generally used on the tested farms and the zoonotic potential of the disease. The phenotypic resistance or intermediate susceptibility of *C. pseudotuberculosis* to benzylpenicillin and meropenem could therefore reflect the fact that β-lactam antibiotics belonged to the most commonly used group of antibiotics on the tested farms. However, the MIC value for a particular antibiotic may not always predict its resulting clinical efficacy [9]. Moreover, genomic detection of antimicrobial resistance determinants using the program Abricate revealed only the determinants responsible for rifamycin resistance with relatively low identity and coverage with reference sequences in the databases. In general, rifamycin resistance is caused by a mutation in the *rpoB* gene encoding the β subunit of DNA-dependent RNA polymerase [51]. Therefore, the low identity of the reference resistance gene indicates that there is not a mutation leading to rifampin resistance. This is consistent with our results obtained in phenotypic analyses. In contrast, the absence of β-lactam resistance genetic markers in the presence of phenotypic resistance highlights the previously open question of the established breakpoint suitability for interpreting the susceptibility of some *Corynebacterium* species to selected antibiotics [20,52].

### 4.3. Genomic Detection of Virulence Factors and Antimicrobial Resistance Determinants

*C. pseudotuberculosis* strains appear to be very successful in infecting the host organism even if they do not carry any antibiotic resistance genes. To better understand the mechanisms of virulence, we analyzed the virulence factors of all studied strains at the genomic level. The main virulence factors found in all the analyzed genomes were endotoxin phospholipase D (*phoD*) and the following virulence factors responsible for iron uptake: *fagC*, *fagB*, *fagA* and *fagD*, placed downstream of *phoD* [53]; the ciu iron uptake and siderophore biosynthesis system represented by the *ciuABCDE* operon; and the hemin utilization system (*hmuTUV*) involved in the utilization of heme iron. Iron is an essential micronutrient that enables the growth of all bacteria, including pathogenic ones, for which reason withholding iron is one of the host defense mechanisms that prevents the growth of pathogenic bacteria. To overcome iron deficiency and successfully infect the host organism, *C. pseudotuberculosis* is reported to harbor the *dtxR* gene which encodes a homologue of the diphtheria toxin repressor DtxR. The *dtxR* gene is activated by iron-regulated processes that control the expression of genes related to iron homeostasis in corynebacteria [54,55]. In addition, the following adherence factors were found as potential virulence factors in all analyzed genomes: SpaA-type pili *strA* and *spaC*, and SpaH-type pili *spaI* and SpaD-type pili *strC*. However, SpaD-type pili *strB* and *spaD* were found in all genomes of the biovar *ovis*, but in only one genome of the biovar *equi* strain. Pili are proteinaceous filaments known to play a role in bacterial adherence. The gene *spaD* in *C. diphteriae* was observed to be involved in the formation of pili that enable this strain to adhere to human pharyngeal epithelial cells [56]. The *strB* gene was reported to encode collagen-binding protein in *Clostridium difficile,* which is important in early biofilm formation and may be involved in host immune escape mechanisms [57].

### 4.4. Carbohydrate-Active Enzymes

CAZymes were annotated in all analyzed genomes to better understand the pathogenesis and survival of *C. pseudotuberculosis*. We identified glycosyl hydrolases responsible for α-glucan synthesis through the GlgE pathway which was reported in the study by Koliwer-Brandl et al. to be involved in the virulence of *Mycobacterium tuberculosis* [38]. Another aspect of *C. pseudotuberculosis* virulence is the synthesis of a cell envelope that not only protects bacterial cells from adverse environmental effects but also contains corynebacterial mycolates. Mycomembrane lipids of *C. pseudotuberculosis* are reported to have a lethal effect on caprine and murine macrophages [58]. The arabinofuranosyltransferase *aftB*, which participates in cell envelope synthesis, was found in every analyzed genome in this study. The *aftB* is essential in *M. tuberculosis*, and the deletion of *aftB* in *Corynebacterium glutamicum* resulted in viable mutants with a decreased abundance of cell wall-bound mycolic acids [59].

In addition to CAZymes involved in α-glucan synthesis/degradation or cell envelope synthesis, we also identified CDS coding for exo-α-sialidase (GH33) and CP40 protein. Exosialidases catalyze the removal of terminal sialic acid residues from various glycoconjugates to release free sialic acid and are involved in pathogenesis, cellular interactions and bacterial nutrition [60]. Endoglycosidase CP40, together with endotoxin phospholipase D, belong to the important virulence factors of *C. pseudotuberculosis* and the results of proteomic and gene expression analyses confirm their participation in the virulence of the bacterium [56,61]. In an experimental study, CP40 was found to have glycosidase activity on IgG and was capable of varying degrees of glycan chain hydrolysis on horse, sheep and four subclasses of human IgG, with no activity on bovine and goat IgG [43]. Endotoxin phospholipase D and CP40 have therefore received much attention in various studies evaluating them as vaccine candidates [62,63].

## 5. Conclusions

This study combined phenotypic and genomic analysis of *C. pseudotuberculosis* biovar *ovis* strains obtained from natural infections in sheep and goats in the Czech Republic and comparative genome analysis of a total of 156 *C. pseudotuberculosis* biovar *ovis* and biovar *equi* strains from different continents, countries and hosts (including strains sequenced in this study). Consistent with other studies, our results show a highly conserved genome of *C. pseudotuberculosis* leading to clonal-like behavior. This is also valid for strains isolated many years ago. The only exception within *C. pseudotuberculosis* is the biovar *equi*, which differs significantly from the biovar *ovis* at the genomic and phenotypic levels. The larger dataset of 156 genomes and subsequent pan-genome calculation led to the finding that the *C. pseudotuberculosis* pan-genome can be considered essentially closed. In every analyzed genome, genetic determinants for virulence factors important for adherence or iron uptake, as well as the exotoxin phospholipase D, were also detected. The general representation of CAZymes in almost all analyzed *C. pseudotuberculosis* genomes showed the biovar *ovis* to be extremely uniform and the biovar *equi* to be more variable in terms of CAZyme profiles.

This is the first comparative genome analysis in the Czech Republic performed on *C. pseudotuberculosis* strains complemented by analyses of biofilm formation and strain susceptibility to selected disinfectants and antibiotics. The results of comparative and phylogenetic analyses show that the *C. pseudotuberculosis* biovar *ovis* obtained from natural infections in sheep and goats from the Czech Republic has a very similar genome to strains from different countries around the world. All Czech isolates were also able to form biofilm of varying intensity and were sensitive to tested disinfectants and almost all antibiotics (except some representatives of β-lactam antibiotics). However, no genetic markers of resistance to β-lactam antibiotics were detected. The obtained genome sequences of the tested *C. pseudotuberculosis* strains from the Czech Republic and the results of genomic and phenotypic analyses can thus be used for further comparative studies to better understand the infectious mechanisms of *C. pseudotuberculosis*, especially at the genetic level.

## Figures and Tables

**Figure 1 microorganisms-12-00875-f001:**
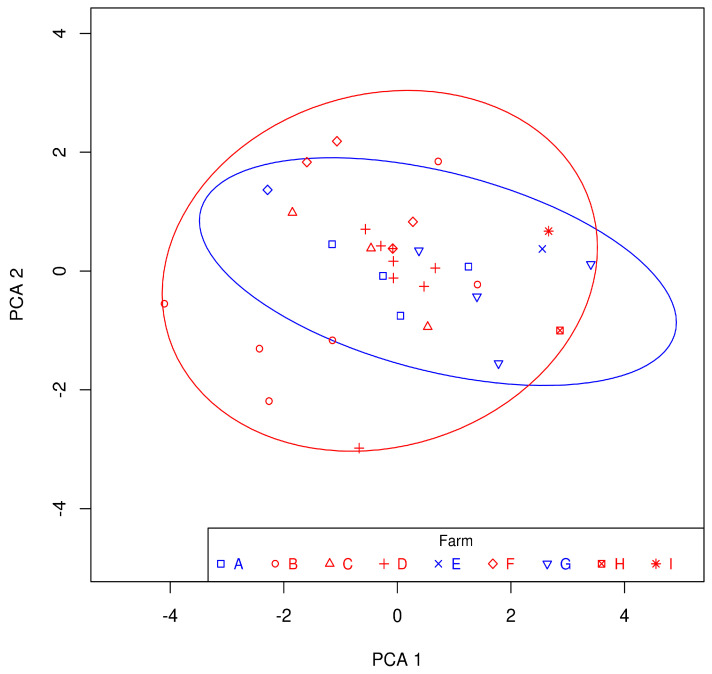
Principal component analysis (PCA) results of log-transformed MBC values determined for the tested disinfectants after their application to *Corynebacterium pseudotuberculosis* biofilms: letters A–I = farm codes; blue color = *C. pseudotuberculosis* strains from goats; red color = *C. pseudotuberculosis* strains from sheep; ellipses = 95% confidence areas for *C. pseudotuberculosis* strains from goats (blue) and from sheep (red).

**Figure 2 microorganisms-12-00875-f002:**
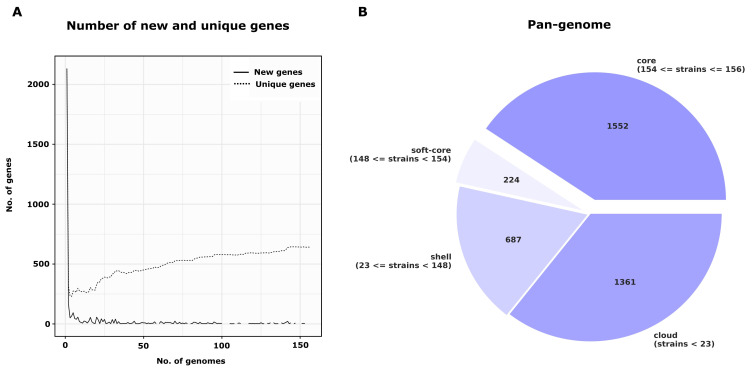
Representation of the pan-genome. (**A**)—change in the number of new and unique genes with the addition of a new genome into the pan-genome (all analyzed *Corynebacterium pseudotuberculosis* of both biovars). (**B**)—number of core, soft-core, cloud and shell genes within the pan-genome (all analyzed genomes of *C. pseudotuberculosis* of both biovars).

**Figure 3 microorganisms-12-00875-f003:**
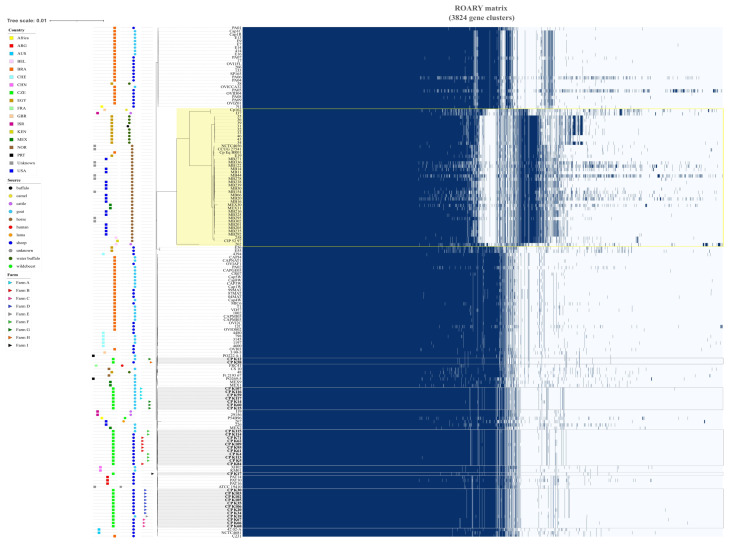
The pan-genome of *Corynebacterium pseudotuberculosis* biovars *ovis* and *equi* in connection with country of origin (squares) and isolation source (circle); strains isolated and sequenced in this study are highlighted with gray rectangles and designated with triangles whose colors correspond to the farm on which the strain was isolated. *C. pseudotuberculosis* biovar *equi* strains are highlighted by a yellow rectangle.

**Figure 4 microorganisms-12-00875-f004:**
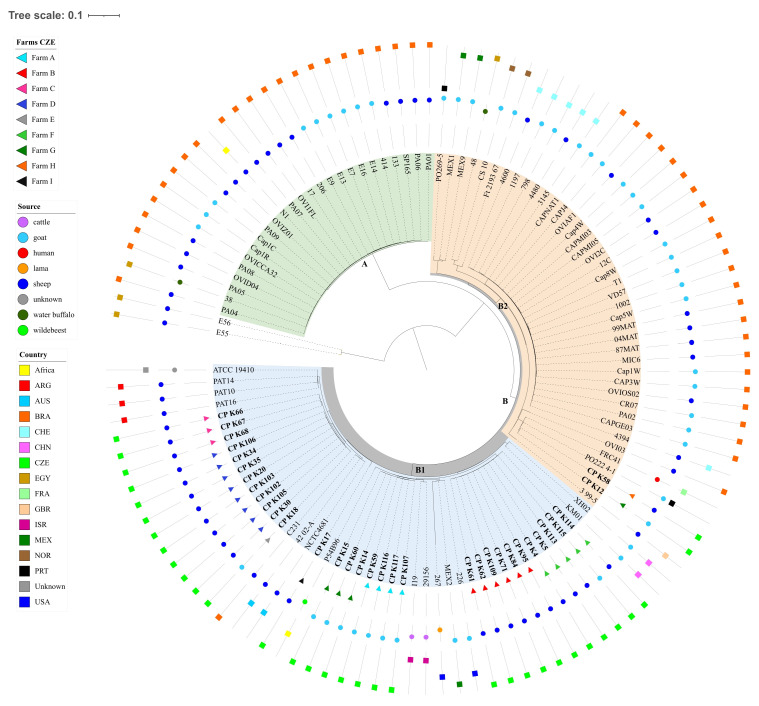
Phylogenetic tree of *Corynebacterium pseudotuberculosis* biovar *ovis* based on core genome SNPs generated by Parsnp and decorated in iTOL. Green coloration highlights clade A, gray color clade B, subclades B1 (light blue) and B2 (light orange). Squares designate the countries of strain origin, circles mark the strain isolation source and triangles mark the farm from which the strains sequenced in this study were isolated.

**Figure 5 microorganisms-12-00875-f005:**
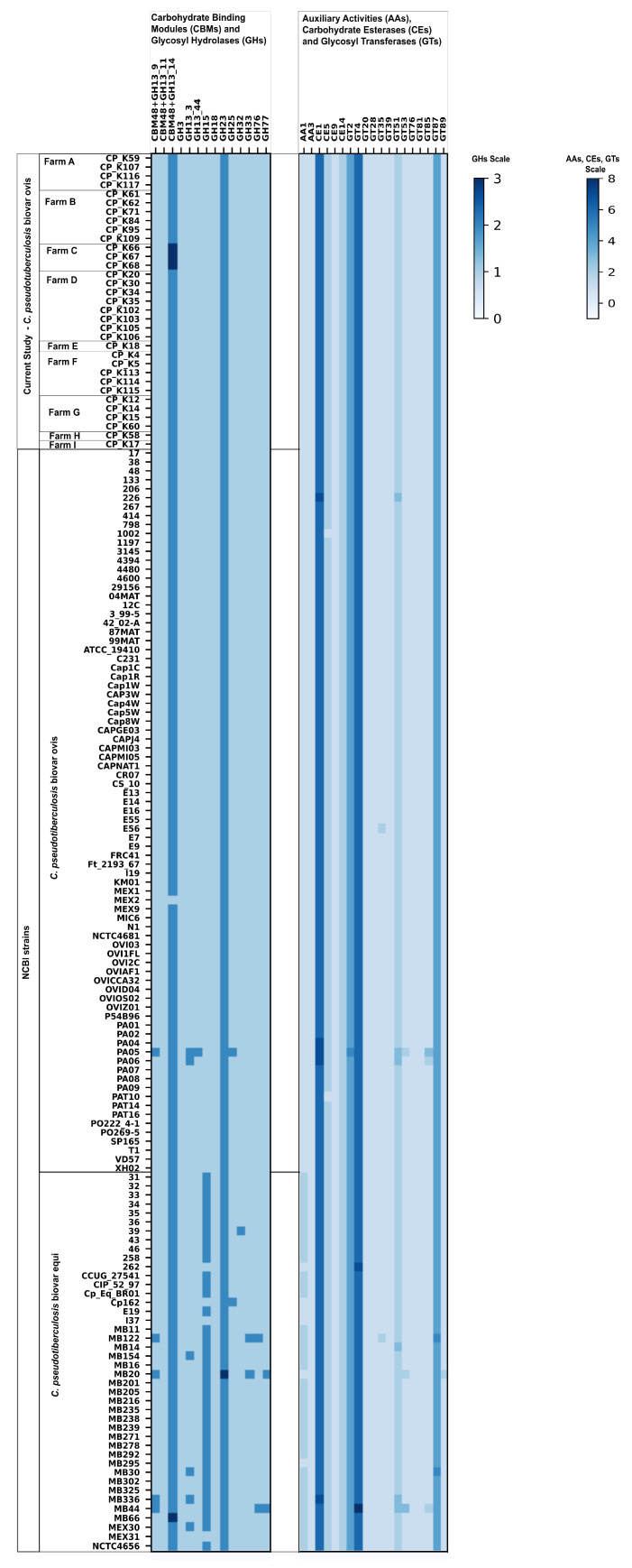
Distribution of CAZymes within *Corynebacterium pseudotuberculosis* biovar *ovis* and biovar *equi* genomes; an underscore between GH names represents a GH subfamily.

**Table 1 microorganisms-12-00875-t001:** Distribution of minimum bactericidal concentration (MBC) for selected disinfectants determined after their application to *Corynebacterium pseudotuberculosis* biofilms.

Disinfectant	MBC (%) (V/V)											RC * (%)
	Number of *C. pseudotuberculosis* Strains with Inhibited Growth											
Benzalkonium chloride	0.0001	0.0002	0.0004	0.0008	0.0016	0.0031	0.0063	0.0125	0.025	0.05	0.1	1–2
	0	0	0	0	0	6	4	13	5	2	3	
Sodium hypochlorite	0.001	0.002	0.004	0.008	0.016	0.031	0.063	0.125	0.25	0.5	1	0.85
	0	0	0	4	12	9	4	3	1	0	0	
Peracetic acid	0.001	0.002	0.004	0.008	0.016	0.031	0.063	0.125	0.25	0.5	1	0.3–1
	0	5	1	1	3	9	6	5	1	2	0	
Chlorhexidine digluconate	0.002	0.004	0.008	0.016	0.031	0.063	0.125	0.25	0.5	1	2	2–4
	0	0	0	2	5	13	9	1	3	0	0	
PVP **	0.010	0.020	0.039	0.078	0.156	0.313	0.625	1.25	2.5	5	10	7.5;
	0	0	0	2	5	2	5	8	8	2	1	10 ***
Ethanol	0.068	0.137	0.273	0.547	1.094	2.188	4.375	8.75	17	35	70	70
	0	0	0	0	0	0	0	2	4	21	6	

* RC = concentration (V/V) recommended by the manufacturer of the disinfectant containing the tested disinfectant agent. ** PVP = polyvinylpyrrolidone iodine. *** can be diluted to 1 and 0.1%.

**Table 2 microorganisms-12-00875-t002:** Distribution of minimum inhibitory concentration (MIC) in *Corynebacterium pseudotuberculosis* strains from goat and sheep farms in the Czech Republic. Interpretation criteria according to the EUCAST clinical breakpoint for *Corynebacterium* spp. [19] or, if absent, according to the CLSI [20]. The cut-off value for trimethoprim/sulfamethoxazole (* Trim./Sulf.) is not known. Gray zones represent values higher than the cut-off values for *Corynebacterium* spp.

Antibiotic Range (mg/L)	MIC Values (mg/L)														EUCAST		CLSI		
	0.008	0.016	0.031	0.063	0.125	0.25	0.5	1	2	4	8	16	32	64	S≤	R>	S≤	I	R>
Benzylpenicillin (≤0.016–≥16)	-	0	0	0	9	6	18	0	0	0	0	0	-	-	0.125	0.125	0.12	0.25–2	4
Vancomycin (≤0.032–≥32)	-	-	0	0	0	0	0	33	0	0	0	0	0	-	2	2	2	-	-
Erythromycin (≤0.008–≥16)	0	1	30	2	0	0	0	0	0	0	0	0	-	-	-	-	0.5	1	2
Clindamycin (≤0.016–≥16)	-	0	0	0	18	15	0	0	0	0	0	0	-	-	0.5	0.5	0.5	1–2	4
Linezolid (≤0.032–≥32)	-	-	0	0	0	0	4	29	0	0	0	0	0	-	2	2	2	-	-
Rifampicin (≤0.032–≥64)	-	-	33<	0	0	0	0	0	0	0	0	0	0	0	0.06	0.5	1	2	4
* Trim./Sulf. (≤0.03/0.59–≥4/76)	0	0	1	12	0	18	2	0	0	0	-	-	-	-	-	-	-	-	-
Meropenem (≤0.016–≥16)	-	0	1	2	8	12	10	0	0	0	0	0	-	-	-	-	0.25	0.5	1

S—susceptible; I—intermediate; R—resistant.

## Data Availability

Sequenced genomes of *Corynebacterium pseudotuberculosis* strains from the Czech Republic were deposited in the NCBI database under Bioproject accession number PRJNA1044565. All remaining data supporting the findings of this study are available in the article and/or Appendix A.

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
