# Peer review of "Ovine and Caprine Strains of Corynebacterium pseudotuberculosis on Czech Farms—A Comparative Study"

_microorganisms, 2024, doi:10.3390/microorganisms12050875_

Round 1

Reviewer 1 Report

Comments and Suggestions for Authors

Dear authors,

Suggestions and comments follow:

Line 54-71: This paragraph contains a series of data without any citations. Furthermore, part of it is a description of methodology, being in an inappropriate place. I ask you to rewrite this last paragraph, using the following structure: 1) paragraph with data from other authors and statements with appropriate citations; 2) separation of objectives in a new paragraph at the end of the introduction; 3) remove the part that describes the methodology of your study, compiling the data in an appropriate section.

Line 188; 194; 227; 306; 411; 430: Abbreviates Genus Corynebacterium.

Line 233; 237; 282; 301; 305; 306; 308; 311; 316; 317; 318; 325; 334; 348; 356; 370; 380; 393; 409; 411; 430; 454; 457; 461; 463; 469: "C. pseudotuberculosis" should be in italics.

In the legend of Figure 1, Figure 2, Figure 4, Figure 5 and Table 2, the name of the bacteria must be complete upon first mention.

There are two Figures 2.

Line 349; 356; 370; 380; 457; 459; 462; 463; 464; 467; 469; 470: equi and ovis should be in italics.

Figure 4 is not clearly displayed.

Kind regards

Author Response

Point 1: Line 54-71: This paragraph contains a series of data without any citations. Furthermore, part of it is a description of methodology, being in an inappropriate place. I ask you to rewrite this last paragraph, using the following structure: 1) paragraph with data from other authors and statements with appropriate citations; 2) separation of objectives in a new paragraph at the end of the introduction; 3) remove the part that describes the methodology of your study, compiling the data in an appropriate section.

Response 1: Thank you for the recommendations. 1) The paragraph does not contain any citations as these are general statements based on the nature of these types of analyses and the purposes for which they are used; 2 and 3) It has been changed.

Point 2: Line 188; 194; 227; 306; 411; 430: Abbreviates Genus Corynebacterium.

Response 2: It has been abbreviated.  

Point 3: Line 233; 237; 282; 301; 305; 306; 308; 311; 316; 317; 318; 325; 334; 348; 356; 370; 380; 393; 409; 411; 430; 454; 457; 461; 463; 469: "C. pseudotuberculosis" should be in italics.

Response 3: It has been changed. This is surprising to us because we had C. pseudotuberculosis in italics in the original uploaded manuscript (as well as biovar ovis and equi). Apparently, there were some changes after the editors formatted the text.

Point 4: In the legend of Figure 1, Figure 2, Figure 4, Figure 5 and Table 2, the name of the bacteria must be complete upon first mention.

Response 4:. It has been changed.

Point 5: There are two Figures 2.

Response 5: It has been changed.

Point 6: Line 349; 356; 370; 380; 457; 459; 462; 463; 464; 467; 469; 470: equi and ovis should be in italics.

Response 6: It has been changed.

Point 7: Figure 4 is not clearly displayed.

Response 7: Thank you for your comment. Figure 4 cannot be displayed at full readable size, it would be too large for a manuscript in this format. Figure 4 has been edited so that when the document is enlarged to a larger size, everything is readable. We also expect an online open access version where the details will be clearly visible.

Reviewer 2 Report

Comments and Suggestions for Authors

The authors have characterized in great detail Corynebacterium strains from sheep and goat farms in the Czech Republic.

The work merits publication, but there are various shortcomings in the manuscript, which should be addressed.

1.      The manuscript is too long for this type of manuscript. It should be reduced in size significantly and should become half its current length. All the methods that were followed are well-established, so the authors should only mention rreferences to previously published technical papers or at the most they should describe techniques in 3 lines. This will suffice, as the reader goes through extensive verbosity in the M & M section.

2.      The novelty in the manuscript is not defined strongly and is not presented well. This must be corrected.

3.      Please explain correctly all the controls that were used in this study. Please underline positive and negative controls, for everything, farms, animals, strains, ambient conditions, everything please. This is very important to correct.

4.      Comment about tables. Most of the tables do not offer much in the main text and can be moved to supplementary material. This will also contribute to reducing the length of the revised manuscript.

5.      Figures. These are OK.

6.      References. Please present references about similar words for all countries of Europe. At the moment, the relevant information is not correct and many references are missing. Please correct.

7.      The Discussion must be divided into 2 or 3 sub-sections for easier flow of reading.

8.      The Conclusions do not align fully with the results. There are some exaggerations, so please tone down the text.

Overall. The manuscript provides useful information, but it must be improved by taking into account the above comments.

Author Response

Point 1: The manuscript is too long for this type of manuscript. It should be reduced in size significantly and should become half its current length. All the methods that were followed are well-established, so the authors should only mention references to previously published technical papers or at the most they should describe techniques in 3 lines. This will suffice, as the reader goes through extensive verbosity in the M & M section.

Response 1: Thank you very much for your comments. We have shortened the text. Unfortunately, it was not possible to reduce the length of the manuscript as significantly as requested. This would have resulted in either the loss of some important data or the procedures of some analyses that we have been modifying in our department over several years of incremental testing and refinement of procedures.

Point 2: The novelty in the manuscript is not defined strongly and is not presented well. This must be corrected.

Response 2:. It has been corrected.

Point 3: Please explain correctly all the controls that were used in this study. Please underline positive and negative controls, for everything, farms, animals, strains, ambient conditions, everything please. This is very important to correct.

Response 3: Positive and negative controls have been specified.

Point 4: Comment about tables. Most of the tables do not offer much in the main text and can be moved to supplementary material. This will also contribute to reducing the length of the revised manuscript. 

Response 4: Some tables have been moved to supplementary materials.

Point 5: Figures. These are OK.

Point 6: References. Please present references about similar words for all countries of Europe. At the moment, the relevant information is not correct and many references are missing. Please correct.

Response 6: We apologize, but we do not fully understand what we should correct? For comparative genomic and phylogenetic analysis, we selected sequences that were suitable for this type of analysis in terms of data quality, which may have excluded some European countries.

Point 7: The Discussion must be divided into 2 or 3 sub-sections for easier flow of reading.

Response 7: It has been divided.

Point 8: The Conclusions do not align fully with the results. There are some exaggerations, so please tone down the text.

Response 8: It has been changed.

Round 2

Reviewer 2 Report

Comments and Suggestions for Authors

The authors have made extensive and careful changes in the manuscript, which is now to publication standard. I have no further comments.